# Design, Manufacturing, and Evaluation of Race and Automotive Prototypal Components Fabricated with Modified Carbon Fibres and Resin

**DOI:** 10.3390/polym16142062

**Published:** 2024-07-19

**Authors:** Dionisis Semitekolos, Andreia Araújo, Raquel M. Santos, Chiara Pernechele, Francesco Panozzo, Luca Vescovi, Costas Charitidis

**Affiliations:** 1Research Lab of Advanced, Composite, Nano-Materials and Nanotechnology (R-NanoLab), School of Chemical Engineering, National Technical University of Athens, 9 Heroon Polytechnique, GR-15773 Athens, Greece; 2Materials and Composite Structures Unit, Institute of Science and Innovation in Mechanical and Industrial Engineering (INEGI), 4200-465 Porto, Portugal; 3Associate Laboratory of Energy, Transports and Aeronautics (LAETA), 4200-465 Porto, Portugal; 4Dallara Automobili S.p.A, Varano De Melegari, 43040 Parma, Italy

**Keywords:** modified resin, electropolymerised CFs, surface treatments, CFRPs

## Abstract

This study explores the enhancement of Carbon Fibre Reinforced Polymers (CFRPs) for automotive applications through the integration of modified carbon fibres (CF) and epoxy matrices. The research emphasizes the use of block copolymers (BCPs) and electropolymerisation techniques to improve mechanical properties and interfacial adhesion. Incorporating 2.5 wt.% D51N BCPs in the epoxy matrix led to a 64% increase in tensile strength and a 51.4% improvement in interlaminar fracture toughness. The electropolymerisation of CFs further enhanced interlaminar shear strength by 23.2%, reflecting a substantial enhancement in fibre–matrix interaction. A novel out-of-autoclave manufacturing process for an energy absorber prototype was developed, achieving significant reductions in production time and cost while maintaining performance. Compression tests demonstrated that the modified materials attained an energy absorption rate of 93.3 J/mm, comparable to traditional materials. These results suggest that the advanced materials and manufacturing processes presented in this study are promising for the development of lightweight, high-strength automotive components, meeting rigorous performance and safety standards. This research highlights the potential of these innovations to contribute significantly to the advancement of materials used in the automotive industry.

## 1. Introduction

The automotive industry has strongly embraced carbon fibres (CF) for its ability to provide durable, and lightweight components, without compromising performance or safety [1]. In fact, lightweight materials offer significant potential to enhance vehicle efficiency and fuel saving, as lighter items require less energy for acceleration [2,3]. Furthermore, components, such as body panels, wheels, rims, and interior finishes benefit from carbon fibre’s unique qualities and pleasing aesthetics. However, while aftermarket CF composite parts are available, they are predominantly featured in luxury sports vehicles and racing cars, particularly due to their high cost when compared to other standard materials. Nevertheless, this paradigm is changing, as can be seen by the automotive CF market, which was valued in 2022 at USD 24.13 billion and was projected to reach USD 64.05 billion by 2032, with a Compound Annual Growth Rate (CAGR) of 11.60% during the forecast period of 2023–2032 [4]. It is also important to state that leading CF providers respond to the demands of vehicle Original Equipment Manufacturers (OEMs), system suppliers, and consumers to create lighter, cleaner, safer, and more cost-effective vehicles. Carbon fibre’s adoption as a replacement for conventional metals and alloys promotes compliance with stringent emission regulations and rising fuel prices, improving engine performance and fuel efficiency. Moreover, carbon fibre’s physical strength surpasses that of common metals, making it a promising material to extend the range of electric vehicles as global demand rises [5,6].

Unlocking CF’s full potential in automotive applications strongly benefits from ongoing research focusing on enhancing the properties of Carbon Fibre Reinforced Polymers (CFRPs). In our study, CFRP modification is achieved by incorporating block copolymers (BCPs) into the thermoset matrix and by anodically oxidizing and polymer grafting the carbon fibre surface. The incorporation of BCPs aims to enhance fracture resistance and improve the interfacial properties, ensuring an efficient load transfer from the matrix to the fibres. Meanwhile, the anodic oxidation process creates functional groups on the carbon fibre surface, which facilitates subsequent polymer grafting with methacrylic acid, leading to improved covalent bonding with the matrix. These modifications are designed to enhance the overall mechanical properties and durability of the CFRP, making it particularly suitable for energy absorbers in automotive applications, which is the main application field of this study.

Despite their promising features, CFRPs are also known to present poor through-thickness properties, mainly associated with the brittle nature of the thermoset polymer base [7]. Furthermore, since the performance of CFRPs is strongly dependent on the fibre–matrix interface, the improvement of the interfacial properties is crucial to ensure an efficient load transfer from the matrix to the reinforcements, aiming to reduce stress concentrations and enhance the overall mechanical properties [8]. One promising approach involves the incorporation of block copolymers (BCPs) into the thermoset matrix, which are polymers comprised of two or more distinct polymer chains covalently connected at their endpoints to form one chain [9,10]. Specifically, AB-type block copolymers consist of two distinct polymer blocks (A and B) connected end-to-end. ABA-type block copolymers have three blocks arranged in a sequence where a central B block is flanked by two A blocks. ABC-type block copolymers have three different polymer blocks in a sequence. These structures can enhance fracture resistance while maintaining essential properties, including glass transition temperature and modulus. The presence of ‘epoxy-miscible’ and ‘epoxy-immiscible’ blocks promotes the self-assembling of BCPs into nanosized structures within the matrix, such as vesicles, spherical micelles, or wormlike micelles, improving mechanical integrity and durability at concentrations below 5 wt.% [11,12].

In addition to the advancements in functionalized polymeric matrices, modifying the fibre surface is another approach for enhancing carbon fibre properties and final CFRP composite components. This involves altering physical, chemical, or morphological properties through techniques such as surface oxidation, roughening, or coating with nanomaterials (carbon nanotubes/graphene oxide, CNTs/GO), plasma modification, and polymer grafting [13,14]. This work focuses on a two-step process involving anodic oxidation and polymer grafting. Anodic oxidation activates the carbon fibre surface, creating functional groups that facilitate subsequent electropolymerisation of methacrylic acid [15]. The polymer’s functional groups promote covalent bonding with the matrix, improving interfacial properties while serving as a sizing agent, streamlining the fibre manufacturing process [16].

The properties of composites resulting from these approaches, namely the incorporation of BCPs into polymer matrices and electropolymerised fabrics, are evaluated in this study, aiming to achieve synergetic effects, through morphological and mechanical testing. These modified materials are then utilized to produce an energy absorber prototype. Energy absorbers are critical components in automotive safety systems, designed to absorb and dissipate energy during impacts, thereby protecting passengers and vehicle structures. These components are often utilized in crash management systems, such as bumper beams and side-impact bars. Traditional manufacturing of CFRP energy absorbers typically involves autoclave processes, which provide high-quality and reliable results but are associated with high costs, long cure times, and limitations in scaling up for mass production.

In this study, we introduce an out-of-autoclave process for producing CFRP energy absorbers. This novel manufacturing methodology offers a promising alternative to conventional autoclave methods by reducing cure time and costs. The process involves the use of a vacuum bench for automated final shaping, allowing for more efficient and scalable production. Our approach not only maintains the high-performance characteristics required for energy absorbers but also aligns with the automotive industry’s push towards more cost-effective and scalable manufacturing solutions for advanced materials.

## 2. Materials and Methods

### 2.1. Materials

Surface treatments (electropolymerisation) have been performed on commercial carbon fibres (T700S C12K-50C PP—Toray, France) with double distilled methacrylic acid (Acros Organics, Waltham, MA, USA), under reduced pressure. For CF pre-impregnation purposes, a three-part resin system (epoxy Araldite LY 556, accelerators 1571/1573, hardener XB 3430—HUNTSMAN Industries, Arlington, TX, USA) was used. For resin modification, a block copolymer (BCP) D51N was purchased from Arkema (Colombes, France). D51N is an ABA-type triblock copolymer, poly(methylmethacrylate-block-butylacrylate-block-methylmethacrylate)—(PMMA-b-PbuA-b-PMMA), (also known as MAM), where the PMMA blocks are miscible with the epoxy resin, while the PbuA blocks are immiscible.

Treated fibres with modified resin were used for the manufacturing of the energy absorber prototype. In particular, these demonstrators and materials used for their production are listed in Table 1. The reference demonstrator is made of alternating plies of commercial twill carbon fabric with DT120 epoxy system, provided by Deltatech (Altopascio, Italy), Toray group, and unidirectional (UD) prepregs made with unmodified epoxy resin. This part is made to have a comparison baseline with the modified prototype. The modified demonstrator is made of alternating plies of commercial carbon fabric prepregs, and prepregs produced using the treated (electropolymerised) UD fibres/modified resin with BCP. The commercial material is necessary to ensure the correct position of the unidirectional layers both during the lamination and curing phases.

### 2.2. Electropolymerisation of CFs

A continuous surface treatment line for fibre treatment has been designed and manufactured based on a two-stage process that has been developed recently in the lab and reported in our previous work; electropolymerisation of MAA [15]. The aim of this technology is to create an interlayer between the fibre and the matrix that improves their adhesion force, resulting in composites with enhanced interlaminar shear strength, a property linked to delamination, which is one of the most common failure mechanisms in composites. The line, which is depicted in Figure 1, consists of the let of tension creel (station 1), where the carbon fibre spool is unwinded and guided through rollers in the solution bath (station 2). Electrodes (l:1000 × w:5 × t:2 mm) dipped in the electrochemical cell are connected to a DC power source that permits at the 1st step, anodic oxidation and, on the 2nd run, enables the successful electropolymerisation of MAA. Fibres are looped into the bath for a total treatment time of 20 min, capable of creating a uniform coating on the CF surface and, afterwards, washed to remove solvent through 2 aqueous baths (station 3). Fibres’ path passes through a nip-roller consisting of 2 cylinders that, under compressed air, squeeze the fibre, removing any excess solution, followed by a vertical furnace that evaporates the remaining solvent (station 4). A feed roller system consisting of 3 cylinders set the speed line (station 5). The final part of the line is the take-up winder that, besides winding the fibres on a new spool, also sets the tension of the system (station 6). The production rate is 24 m per hour with the current residence time in the bath. The electropolymerisation takes place at −0.435 V in an aqueous solution containing 0.3 M acrylic acid, 0.4 M ZnCl as the supporting electrolyte, and 10 mM N, N-Methylene-bis(acrylamide), which acted as a crosslinker for the grafted polymer as described in our previous work [15].

CF morphology of reference and after electropolymerisation treatments were evaluated using Scanning Electron Microscopy (SEM) analysis with a Hitachi Tabletop Microscope TM3030 Scanning Electron Microscope (Hitachi, Tokyo, Japan) equipped with an Energy Dispersive X-Ray Spectrophotometer (EDS) system (QUANTAX 70). The thermal stability was characterised by Thermogravimetric Analysis (TGA) using an STA 449 F5 Jupiter (Munchen, Germany). A temperature sweep from 100 to 700 °C was conducted at a heating rate of 20 °C/min, under nitrogen flow (50 mL/min). Fourier transform infrared (FTIR) spectra were recorded with a Cary 630 spectrometer (Agilent, Santa Clara, CA, USA) with an Attenuated Total Reflectance (ATR). The samples were used directly without further preparation. The IR spectra were recorded at a 4 cm^−1^ spectral resolution over the range from 400 to 4000 cm^−1^ with background subtraction.

### 2.3. Prepreg and Coupon Manufacturing

The incorporation of D51N into the epoxy matrix was performed by mechanical mixing and stirring of both components at 150 °C, ensuring their complete dissolution. After the addition of hardeners, unmodified and modified resins were poured into silicon moulds and cured for 2 h at 120 °C and post-cured for 2 h at 180 °C. Prepreg materials using commercial or treated fibres, and unmodified or modified resins with D51N, were produced using a Drumwinder, Century Design, and the fibre volume fraction (VCF) was determined according to ASTM D3529 [17]. Before prototype construction, UD-CFRP laminates containing from 12 to 48 layers, depending on the desired thicknesses, were consolidated in an autoclave for 2 h at 120 °C and 3.5 bar. Table 2 presents the V_CF_ of each CFRP panel produced.

#### Material Testing

The mechanical characterisation was performed using an INSTRON 5900R universal instrument (Norwood, MA, USA) with a cell of 5 or 200 kN on at least 5 specimens for each formulation and type of test. Tensile tests of the nanocomposites were obtained from dog-bone specimens according to ISO 527-2 [18], aiming to determine the optimal content of D51N to be used in modified prepreg materials. Fibre-dominated failure modes were evaluated using tensile tests at 0° according to ASTM 3039 [19], and interlaminar shear strength (ILSS) properties of the CFRPs containing sized CFs were determined by ASTM D2344 [20]. Specimens for Mode I interlaminar fracture toughness (GIC) of the CFRPs containing modified resin with D51N were prepared according to our previous work [8] and according to ASTM D5528 [21]. G_IC_ was determined using a modified beam theory (compliance-based beam method) [8].

Dynamic mechanical analysis (DMA) was conducted using a TA Instruments Q800 (New Castle, DE, USA) in single cantilever mode at 1 Hz of frequency. The temperature varied from 30 to 200 °C at a 5 °C/min rate and the glass transition temperature (Tg) was determined at the peak value of tan δ. SEM images were obtained in a Nova 200 NanoLab from FEI Company (Hillsboro, OR, USA).

### 2.4. Energy Absorber Design

An energy absorber (automotive part) was designed to evaluate the feasibility of the novel materials. This component is part of the vehicle structure, which is designed to absorb kinetic energy deriving from a crash through plastic deformation. Traditional energy absorbers are typically comprised of aluminium in the form of extruded bars. CFRP composites are excellent candidates for the manufacturing of energy absorbers, since the light weight of this class of materials, combined with the outstanding specific energy absorption (80 J/g vs. 40 J/g of aluminium alloys) can lead to extremely high-performance products [22,23]. However, the time-demanding processes that are typically necessary for these materials made CFRP energy absorbers only suitable for high-level sports cars.

To overcome some of these challenges, an energy absorber suitable for the automotive market was designed from the initial stages with the aim of introducing an innovative manufacturing methodology that can significantly reduce time and production costs. Therefore, compression moulding and automatic vacuum-assisted preforming were introduced for prototype manufacturing.

The design process was supported by Finite Element Method (FEM) analysis, aimed at determining the optimum layup to satisfy the mechanical requirements. In particular, Dallara set targets are as follows:

The structure shall absorb the axial impact of a mass of 780 kg at a speed of 11 m/s (47.19 kJ);The peak deceleration shall not exceed 20 g.

FEM simulations were carried out considering a standard carbon fibre prepreg epoxy system, resulting in a layup with zones having differential thicknesses (Figure 2) to gradually absorb the impact energy.

A more detailed view of the plies can be seen in Figure 3, where the lines represent the perimeter of the reinforcing patches of the part. In more detail, red lines represent +-45° plies whereas green lines represent 0°/90 plies.

The electropolymerised CF fabric developed in this work was placed in the yellow area of Figure 4 and Figure 5 (across all absorber sections A–C) with the aim of maximizing the influence of the reinforcement while at the same time maintaining a simple stacking sequence.

Detailed information on the lamination sequence can be seen in Figure 6, where the exact lamination phases and orientations of the reinforcements are listed.

As can be seen in the lamination table (Table 3), 1 layer of commercial CFRP fabric alternated with 4 layers of modified 0/90 UD.

### 2.5. Manufacturing of Energy Absorber

Prepreg material was cut according to the CAD specifications. In the first trials of the prototype manufacturing, prepreg layers were manually laminated into a resin pre-forming mould (Figure 7a), whereas in the final trials, a semi-automatic pre-forming bench was manufactured (Figure 7b,c). A pressure of 0.8 bar was applied for 2 min to guarantee compact plies.

Left and right preforms were combined and a silicone bag was placed inside. The preforms were placed in the mould, which was then closed to allow curing of the part at 125 °C. A pressure of 2 bar was applied to the silicon bag for the first 5 min and then a pressure of 6 bar was applied for the remaining 25 min. After demoulding, the energy absorber was placed in a trimming tooling where metallic inserts were bonded and holes were drilled (Figure 8).

Energy absorbers were subsequently trimmed with an automatic robot, resulting in the machine version shown in Figure 9. The holes were necessary to constrain the part to a plate in case of compression test and were also necessary as a reference for the insert bonding phase.

The main parameters used for the manufacturing of the energy absorber in its final configuration are listed in Table 4.

The rear absorbing structures of racing cars are typically produced by manually laminating prepreg sheets in open moulds, followed by bagging and curing in an autoclave. This process, known for its high-quality output, is also time-intensive and costly. The entire process, including the heating, dwelling, and cooling phases of the autoclave cycle, typically lasts around 4 h. Additionally, the lamination and preparation steps, including the application of vacuum bags, require approximately 15 h. By contrast, the out-of-autoclave process developed in this work, for the same geometric structure, significantly reduces these times. Utilizing vacuum preforming, the suggestive method requires only 10 h for the lamination phase, and just 70 min for curing and demoulding. This approach not only shortens the production time but also dramatically reduces costs. The out-of-autoclave method lowers utility expenses by almost half compared to the traditional autoclave process, primarily due to the reduced energy consumption during the shorter cure cycle. Additionally, labour costs are cut to about one-third, as the simplified process reduces the manual intervention required for lamination and preparation.

To compare the reference with the energy absorber with modified materials, compression tests were performed. In particular, as testing a full rear impact structure would not have been possible due to energy and reached forces and material quantities, smaller coupons with a similar shape were laminated with the materials of interest and tested statically in a lab-scale mechanical testing machine (analysis on Section 3.3). The mechanical characterisation was conducted using an MTS Insight universal instrument with a 150 kN cell on 3 samples per case.

## 3. Results and Discussion

### 3.1. Characterisation of Modified CFs

#### 3.1.1. SEM

Reference fibres (Figure 10a) exhibit a smooth and striped surface. On the contrary, as can be observed from Figure 10b,c, a polymer film is formed as a coating on the fibre surface, which is related to the electropolymerisation of the MAA monomer. The fibre is initially oxidized to create oxygen-containing groups so that the polymerisation reaction can take place. EDX analysis confirms the increase in oxygen levels from 1.7% on the reference fibre to 12.4% on the electropolymerised fibre (Figure 11). The polymer is applied as a thin film on the majority of the fibres, with rough spots that promote mechanical interlocking with the cured resin.

#### 3.1.2. TGA

TGA was used to study the thermal decomposition of the reference and treated CFs. As highlighted in the temperature ranges of 100–500 °C for the reference (red line) and treated (black line) fibres, there is a steep drop that is usually attributed to decomposition, evaporation, or other chemical reactions (Figure 12). In this case, it is the decomposition of the pre-existing sizing for the reference fibres and the electropolymerised PMMA for treated fibres. As indicated in the technical data sheet (TDS) of T700s reference fibres, the amount of sizing is 1%—this was also confirmed through TGA analysis—while the electropolymerisation results in a higher polymer yield on the fibre surface, reaching 3%.

#### 3.1.3. FTIR

Figure 13 displays the FTIR spectra for untreated and treated CFs. The untreated fibre exhibits several peaks in the mid-infrared range, accompanied by noticeable noise due to the low commercial sizing content (approximately 1%). Comparison of these results with published mid-IR spectra of epoxy resins [15] reveals similar peaks. Specifically, in Figure 13a, the peaks at 1509 cm^−1^ and 800 cm^−1^ correspond to the C-C single bond stretch vibration of the aromatic ring and the C-O-C stretch vibration of the oxirane group, respectively. Additionally, the peaks at 1036 cm^−1^ and 1234 cm^−1^ are associated with the stretching vibrations of C-O-C and C-O of ethers, respectively. In Figure 13b, representing the electropolymerised poly(methacrylic acid) (PMAA) fibres, the spectrum showed significantly reduced noise compared to the untreated fibres. Two additional peaks at 1699 cm^−1^ and 2934 cm^−1^ indicate the presence of carbonyl (C=O) and hydroxyl (O-H) stretch vibrations of the carboxylic acid group, confirming the successful incorporation of PMAA onto the fibre surface.

### 3.2. Coupon Testing

#### 3.2.1. Tests of Modified Resin with BCP

The modification of polymeric matrices is a commonly used strategy for improving the performance of matrix-dominated properties in CFRPs. As previously mentioned, the benefit of adding BCPs resides in their ability to self-assemble in nanosized structures that can significantly increase the fracture resistance of the matrix with minimum impact on glass transition temperature and modulus at concentrations below 5 wt.% [24,25]. After a review of the literature, nanocomposites containing modified resin with 2.5 wt.% D51N were first produced and their thermomechanical and mechanical properties were studied and compared with neat resin. The results are depicted in Figure 14, along with an SEM image of the fractured surface of the modified sample.

As can be observed, the reference material presents a storage modulus (E′) of approximately 2 GPa, which decreases with increasing temperature, in particular above 130 °C. Looking at the tan δ curve, a peak is observed at 165 °C (Tg), which represents an α-transition of the epoxy system related to the mobility of a major chain. With the incorporation of 2.5 wt.% D51N, E′ slightly increases in comparison with the reference, while the Tg shifts to lower temperatures (162 °C). A more accentuated decrease in E′ is also observed at around 100 °C, which denotes the glass transition temperature of the PMMA block [26]. According to Liu et al. [27], these results suggest that D51N has an influence on the epoxy network formation or in the network molecular mobility (plasticization effect), in particular due to the softness of the PbuA block. Nevertheless, there are no significant changes for this particular loading of BCP.

Tensile results showed the positive impact of adding 2.5 wt.% D51N to the epoxy matrix, with a 64% improvement in tensile strength, while Young’s Modulus was 10% higher than that of the reference without BCP incorporation. According to the literature, BCPs are known to improve the fracture toughness (K_I_c) of epoxy while maintaining or slightly decreasing the tensile properties of the modified material [25,28,29]. For instance, Chen et al. [30] produced epoxy-based nanocomposites modified with different loadings (ranging from 2 to 12 wt.%) of three types of MAM triblock copolymers and obtained a decrease in the tensile properties of the materials, while fracture toughness was enhanced. In general, the decrease in the tensile properties is commonly associated with the softer nature of the added blocks, such as PbuA. Nevertheless, Tang et al. [31] also reported tensile strength increases of up to 17 and 31% with the addition of non-reactive and reactive triblock copolymers, respectively, which were lower than those obtained in this work.

SEM observations of the tensile specimens denoted a rough fractured surface, normally associated with improvements in fracture energy dissipation, and the presence of well-dispersed nanospheres or cavities in the range of 300–800 nm. Hence, cavitation growth may be the toughening mechanism responsible for the enhancement of the tensile properties, as previously reported in the literature [25,30].

Based on these results, modified resin with 2.5 wt.% D51N was further used to prepare UD-CFRPs, aiming to obtain multiscale materials with improved interlaminar properties. The interlaminar fracture toughness (G_IC_) of these materials was determined and the results are presented in Table 5.

As can be observed, the toughening of the epoxy matrix using 2.5 wt.% of D51N also enhanced GIC by more than 50% when compared to the unmodified CFRP. This increase may also be explained by the increased affinity between the epoxy matrix and the carbon fibres due to the presence of the BCP, increasing the energy necessary to compromise the fibre–matrix interface and propagate the crack in the test sample. Similar results were obtained by Klingler et al. [26] for CFRPs prepared by hand layup followed by consolidation in an autoclave using UD-CF fabric and modified matrix with BCP D52N (similar triblock copolymer to D51N), with interlaminar fracture toughness improvements of 45% with 2 wt.% BCP addition. Chong et al. [32] also observed a 42% interlaminar improvement in CFRPs made by resin infusion with 2.5 wt.% of a triblock copolymer of poly(styrene)-b-1,4-poly(butadiene)-b-poly(methyl methacrylate), SBM.

#### 3.2.2. Tests with Modified CFs

Comparing the performance of composite coupons fabricated with reference CF fabrics versus those with electropolymerised CF fabrics, several key findings emerge (Table 6). The Ultimate Tensile Strength (UTS) showed a slight increase of approximately 5.9%, while the Modulus displayed a negligible decrease of around 2.3%, and the Flexural Strength was approximately 2.2% higher in the electropolymerised CF fabric panels compared to the reference panels. All these variations fall within the error limits compared to the reference panels. This consistency implies that the modification process does not significantly affect the intrinsic properties of the carbon fibres, as these properties are primarily fibre-oriented and less influenced by interactions with the matrix. Contrarily, the Interlaminar Shear Strength (ILSS) results showed a notable improvement of approximately 23.2% in the electropolymerised CF fabric panels compared to the reference panels. This enhancement suggests that the modification process positively impacts the fibre–matrix interaction. Specifically, the presence of PMMA from the electropolymerisation process likely facilitates stronger chemical bonding between the matrix and the carbon fibres. Consequently, this enhanced bonding mechanism contributes to the substantial increase in ILSS as it directly affects the material’s resistance to interlaminar shear forces. While the modification process does not significantly alter the fibre properties (reflected in UTS, Modulus, and Flexural Strength), it notably enhances the fibre–matrix interaction, leading to a significant improvement in ILSS. These findings highlight the potential of electropolymerised CF fabrics in enhancing the mechanical properties of composite materials, particularly in applications requiring superior interlaminar shear strength, such as automotive and race prototypal components.

### 3.3. Energy Absorber Testing

In order to evaluate the performance during axial crushing of the material, dummy samples representative of the rear absorbing structure were designed and produced using both the reference and modified materials (2.5 wt.% D51N). This step was essential as testing the full laminated structure internally was not feasible due to anticipated high loads. The produced dummy rear impact structure is illustrated in Figure 15, featuring a height of 110.1 mm and a width of 209.5 mm for the crashing structure. Additionally, lateral reinforcement and a base with four fixing holes ensure proper sample positioning in the testing machine. The results are presented in Table 7.

Τhe initial findings indicate promising results as the presence of the modified materials, including electropolymerised CFs and block copolymers (BCPs), does not compromise the performance of the reference material in terms of energy absorption during axial crushing. With an average absorbed energy of 93.7 J/mm for the reference samples and 93.3 J/mm for the modified, the modified materials exhibit comparable performance. However, to further optimize the material’s energy absorption capabilities, exploring different ratios of BCPs and the addition of modified fabrics could be beneficial. By systematically varying these ratios, an optimal combination might be uncovered that maximizes energy absorption without compromising mechanical integrity. These preliminary findings offer a solid foundation for future investigations aimed at refining the composition of the modified materials for enhanced performance.

## 4. Conclusions

This study presents a detailed investigation into the design, manufacturing, and evaluation of race and automotive prototypal components fabricated with modified materials towards enhancing the performance and viability of CFRPs for automotive applications. The incorporation of BCPs into the epoxy matrix has shown promising results in improving fracture resistance while maintaining essential properties. The addition of 2.5 wt.% D51N, an ABA-type triblock copolymer, resulted in a 64% enhancement in tensile strength and a 51.4% increase in interlaminar fracture toughness (GIC) compared to unmodified CFRPs. These findings highlight the potential of BCPs in reinforcing the matrix and enhancing overall mechanical properties. Furthermore, the modification of CF surfaces through electropolymerisation has proven effective in strengthening the fibre–matrix interface. SEM analysis revealed the formation of a polymer film on the fibre surface, facilitating improved adhesion to the matrix. Mechanical testing of composite panels fabricated with electropolymerised CF fabrics exhibited a notable 23.2% increase in ILSS compared to reference panels. The manufacturing process of the energy absorber demonstrates the integration of novel materials and innovative techniques to meet automotive industry demands. The compressive tests conducted on dummy rear impact structures demonstrated comparable specific energy absorption between reference and modified materials, affirming the feasibility of incorporating modified materials into automotive components.

## Figures and Tables

**Figure 1 polymers-16-02062-f001:**
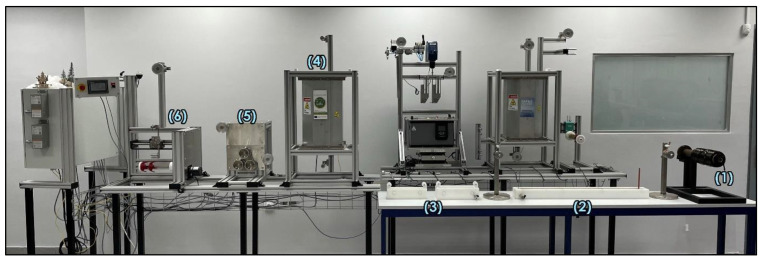
Electropolymerisation line setup.

**Figure 2 polymers-16-02062-f002:**
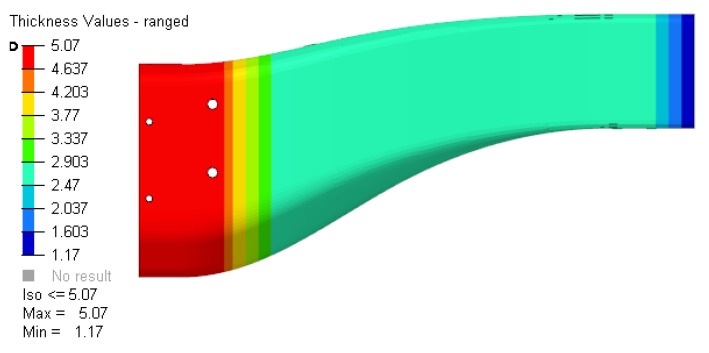
Thickness map of the energy absorber prototype.

**Figure 3 polymers-16-02062-f003:**
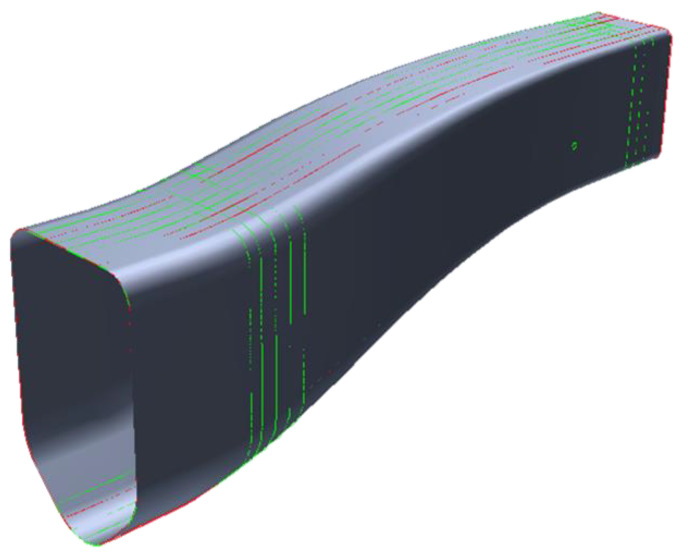
Computer-aided design (CAD) geometry of the energy absorber and overview of the reinforcement patches.

**Figure 4 polymers-16-02062-f004:**
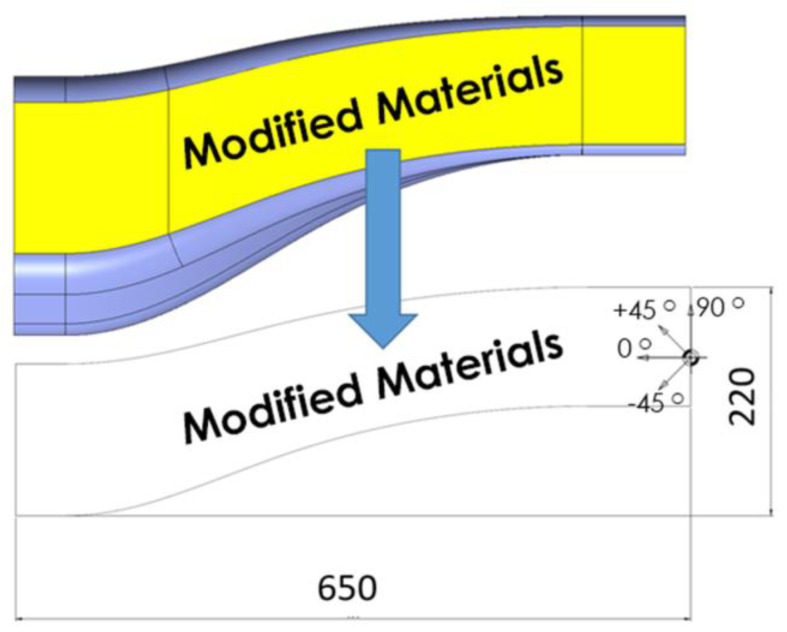
Modified patch placements (dimensions in mm).

**Figure 5 polymers-16-02062-f005:**
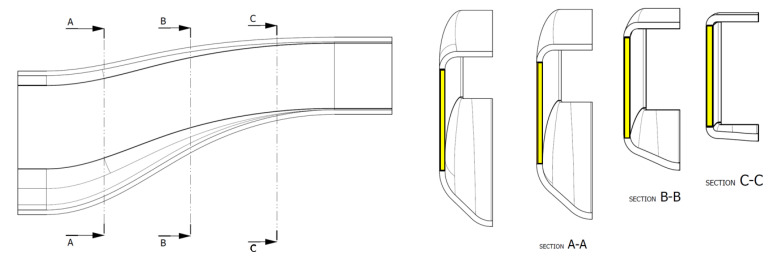
Detail of the lamination patches and sections.

**Figure 6 polymers-16-02062-f006:**
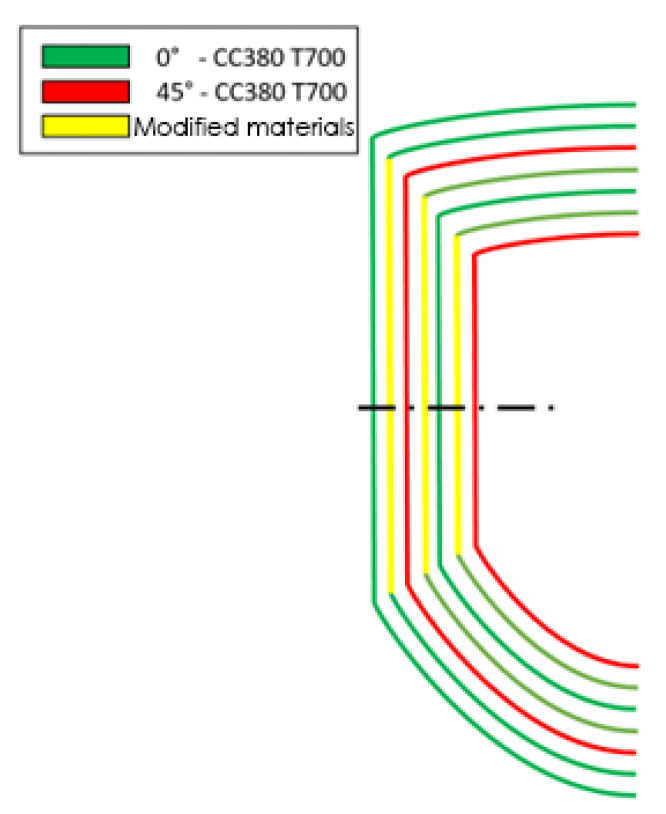
Details of the lamination sequence.

**Figure 7 polymers-16-02062-f007:**
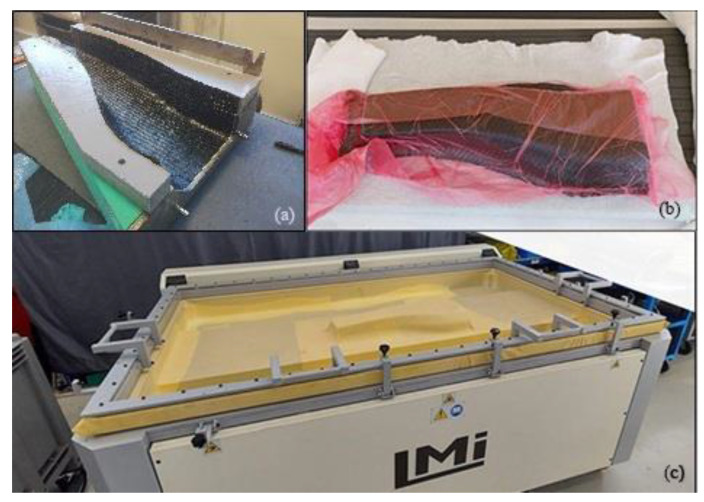
(**a**) Manual preforming tool, (**b**,**c**) semi-automatic preforming through the vacuum bench.

**Figure 8 polymers-16-02062-f008:**
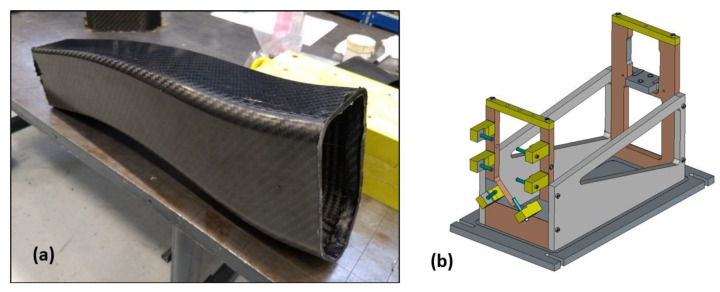
(**a**) Raw energy absorber after demoulding, and (**b**) trimming tooling used.

**Figure 9 polymers-16-02062-f009:**
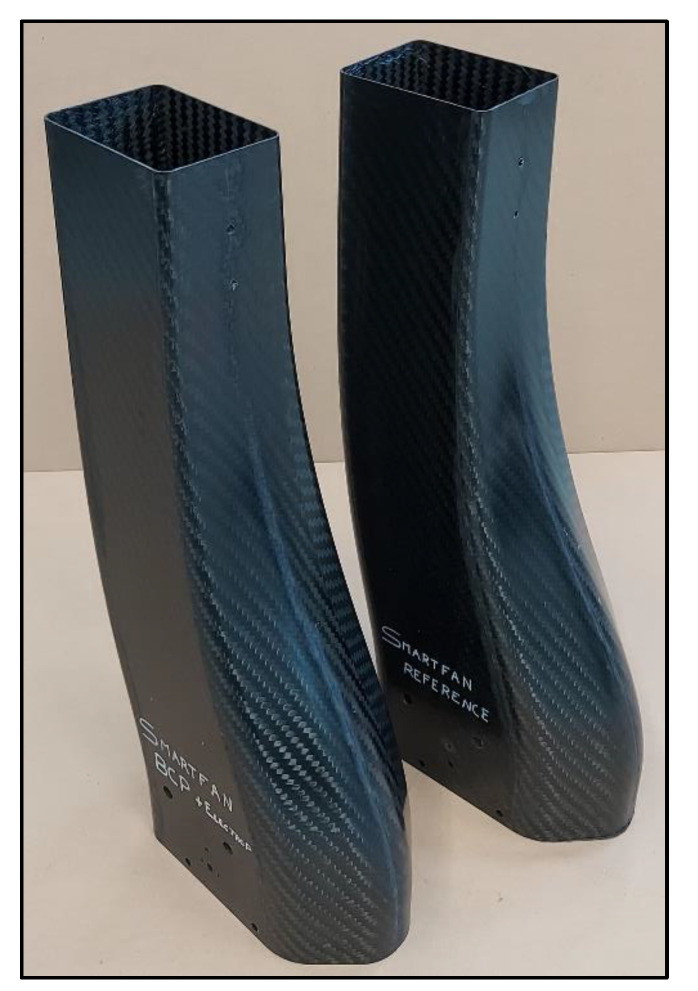
Machined energy absorbers: “Reference” and “Modified”.

**Figure 10 polymers-16-02062-f010:**
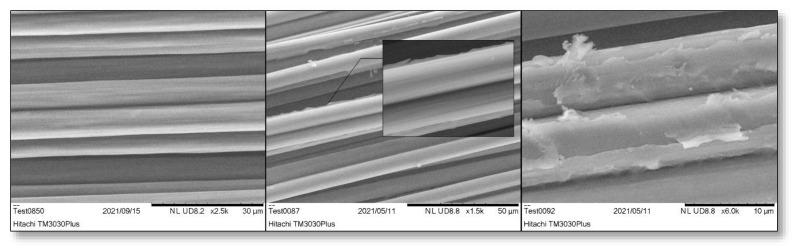
SEM picture of (**a**) commercial CF, (**b**) oxidized CF, and (**c**) electropolymerised CF.

**Figure 11 polymers-16-02062-f011:**
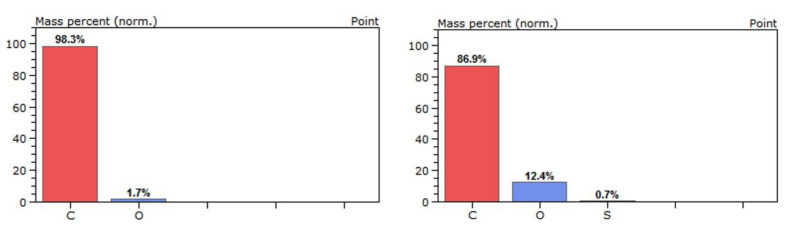
EDX analysis of commercial (**left**) and electropolymerised CF (**right**).

**Figure 12 polymers-16-02062-f012:**
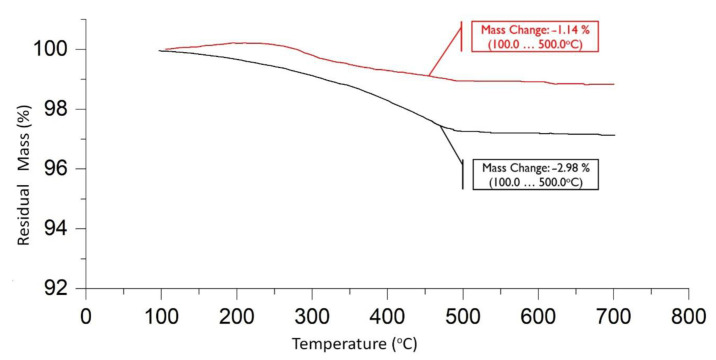
TGA of reference (red line) and treated (black line) CFs.

**Figure 13 polymers-16-02062-f013:**
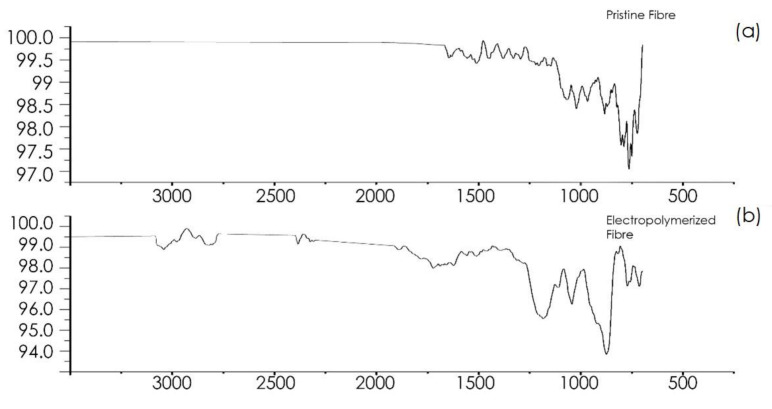
FTIR spectrum of (**a**) pristine and (**b**) electropolymerised fibres.

**Figure 14 polymers-16-02062-f014:**
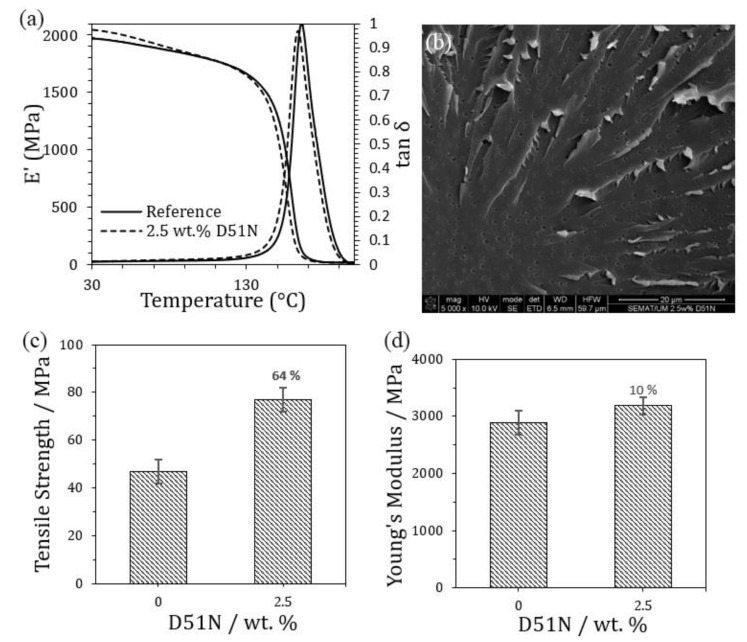
(**a**) representative storage modulus (E′) and tan δ curves obtained from DMA experiments, (**b**) SEM image of the fractured surface of the modified nanocomposite, (**c**,**d**) tensile results comparing the unmodified reference with the nanocomposite containing 2.5 wt.% D51N.

**Figure 15 polymers-16-02062-f015:**
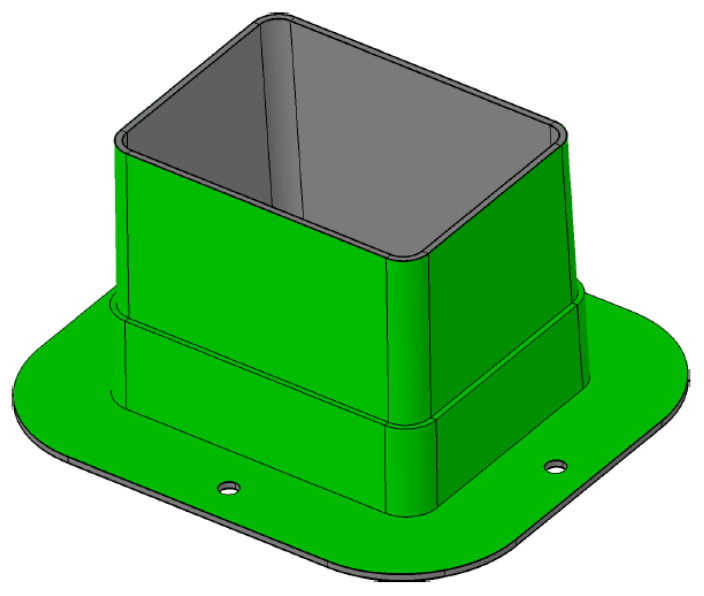
Dummy rear impact structure (RIS).

**Table 1 polymers-16-02062-t001:** Materials used for the production of energy absorber prototypes.

Energy Absorber Type	Fibre	Resin	Amount (m^2^)	Notes
Reference	380 gsm T700 twill 2 × 2	DT120 epoxy	5.0	Commercial Materials
	T700s UD	LY556 epoxy	2.0	Commercial Materials
Modified	380 gsm T700 twill 2 × 2	DT120 epoxy	5.0	Commercial Materials
	T700 electropolymerised UD	LY556 epoxy with BCP	2.0	Modified Materials

**Table 2 polymers-16-02062-t002:** V_CF_ of each CFRP composite panel produced.

Samples	Test	V_CF_ %
Reference for electropolymerised	Tensile 0°	51
	ILSS ^1^	48
Electropolymerised	Tensile 0°	49
	ILSS	47
Reference for D51N	G_IC_	59
2.5 wt.% D51N	G_IC_	58

^1^ Interlaminar shear strength

**Table 3 polymers-16-02062-t003:** Lamination table.

Fase RH	Fase LH	Θ (°)	Materials
F-1010	F-2010	0	CC380 T700
F-1010	F-2020	0	Electropolymerised UD
		90	Electropolymerised UD
		0	Electropolymerised UD
		90	Electropolymerised UD
F-1040	F-2040	45	CC380 T700
F-1060	F-2060	0	Electropolymerised UD
		90	Electropolymerised UD
		0	Electropolymerised UD
		90	Electropolymerised UD
F-1080	F-2080	0	CC380 T700
F-1100	F-2100	0	Electropolymerised UD
		90	Electropolymerised UD
		0	Electropolymerised UD
		90	Electropolymerised UD
F-1130	F-2130	45	CC380 T700

**Table 4 polymers-16-02062-t004:** Parameters for the production of the energy absorber.

Parameter	Value
Preforming pressure	0.8 bar
Preforming duration	2 min
Silicon bag pressure	4 bar for 5 min, then 5 bar
Curing time	30 min
Curing temperature	125 °C

**Table 5 polymers-16-02062-t005:** Interlaminar fracture toughness.

Samples	G_IC,_ N.mm^−1^	Improvement, %
Unmodified CFRP (reference)	0.37 ± 0.02	=
2.5 wt.% D51N	0.56 ± 0.03	51.4

**Table 6 polymers-16-02062-t006:** Comparison of mechanical properties between composite panels fabricated with reference and electropolymerised carbon fibre fabrics.

Property	Reference	Electropolymerised	Difference, %
UTS (MPa)	602.6 ± 32.6	638.2 ± 39.4	+5.9
Modulus (GPa)	6.1 ± 0.8	6 ± 0.5	−2.3
Flexural (MPa)	531.4 ± 62.2	543.1 ± 60.7	+2.2
ILSS (MPa)	28.5 ± 1.9	35.1 ± 2.2	+23.2

**Table 7 polymers-16-02062-t007:** Results of the compression tests.

Dummy Type	Specific Energy Absorption, J/mm
Reference dummy	93.7 ± 3.5
Modified dummy	93.3 ± 4

## Data Availability

Data is contained within the article.

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
