# Peer review of "Design, Manufacturing, and Evaluation of Race and Automotive Prototypal Components Fabricated with Modified Carbon Fibres and Resin"

_polymers, 2024, doi:10.3390/polym16142062_

Round 1

Reviewer 1 Report

Comments and Suggestions for Authors

The study presented the fabrication of CFRP composites with polymer matrix modified by BCPs and CF coated with electropolymerized PMMA. Their structure and mechanical properties were characterized. Improvements are needed before being accepted by the journal. Comments are given as below:

 1. Some experimental details need to be given. Any process for the pretreatment of carbon fibers before the anodic oxidation and electroplating? What is the electrochemical plating condition for the deposition of MMA? How much is the size of the electrodes in the cell and content of PMMA in the modified CFs?

 2. Any tensile strength loss of the carbon fibers after the oxidation treatment?

 3. The figure about the Young's modulus of the composites in Fig 14 is obviously incorrect. The values should not be so large.

 4. Figure caption for Fig 14 is not suitable. The one for the figure about the modulus was not included.

5. How to make the sample for interlaminar fracture toughness and calculate the value? What is the formula used for the calculation? Its unit in Table 5 is different from other work (for instance, Composites Science and Technology 2017, 140, 46).

Comments on the Quality of English Language

 English needs to be improved and some typo could be found. Such as the missing of unit or words in line 142, 307, and 352 etc.

Author Response

Comment 1: Some experimental details need to be given. Any process for the pretreatment of carbon fibers before the anodic oxidation and electroplating? What is the electrochemical plating condition for the deposition of MMA? How much is the size of the electrodes in the cell and content of PMMA in the modified CFs?

Response 1: The requested information has been provided in Section 2.2 with trach changes. This includes information related to the process of electropolymerisation along with its conditions. Not extensive description was added to avoid repeating information that was already provided in our previous publication which was related to the process development. The PMAA is approximately 3% of the fibre total weight as investigated in our previous work with Thermogravimetric analysis. 

Comment 2: Any tensile strength loss of the carbon fibers after the oxidation treatment?

Response 2: The oxidation process has been carefully developed in order to avoid damage on the structural properties (modulus) of fibres. It is a mild treatment that introduces oxygen groups on the fibre surface so the electropolymerisation of MAA can take place. In our 1st work a small increase on bundle tensile strength has been published that was attributed to the improved adhesion of fibre-resin.

Comment 3: The figure about the Young's modulus of the composites in Fig 14 is obviously incorrect. The values should not be so large.

Response 3: Thank you for your comment. Figure has been changed in the document. It was changed to MPa and added a), b), c) and d) to the figure.

Comment 4: Figure caption for Fig 14 is not suitable. The one for the figure about the modulus was not included.

Response 4: Thank you for your comment. The caption has been changed in the document.

Comment 5: How to make the sample for interlaminar fracture toughness and calculate the value? What is the formula used for the calculation? Its unit in Table 5 is different from other work (for instance, Composites Science and Technology 2017, 140, 46).

Response 5: Specimen preparation and calculation of the Mode I test is a standardized procedure and that can be done according to ASTMD5528. It was also detailed reported in our previous work. We included in the manuscript the citation which describes the method to prepare the specimens and calculate the interlaminar fracture toughness.  As for the units, we followed again our work, but it is equivalent to the unit in the paper you mentioned. 1kJ/m2 = 1N/mm

Reviewer 2 Report

Comments and Suggestions for Authors

Dear Authors

Your article mentions about the auto-clave process of a carbon fiber reinforced polymer (CFRP) energy absorber. However, there are some areas that could be improved to provide a clearer understanding of your research:

•CFRP modification: In the introduction, you mention CFRP but don't discuss the specifics of the modifications made.  A clearer explanation of the CFRP used in the energy absorber would be beneficial.

•Meaning of AB, ABA… type: Define the terminology used, such as the meaning of AB, ABA…  on line 62.

•Energy absorber background: The introduction should provide more context on the energy absorber itself, including its applications. How does the autoclave process play a role in the functionality of the energy absorber?

•Line 118-129 & Fig 1 clarity:  If the line setup described and Fig 1 represent a new process, a clear explanation is needed. If it references existing research, please cite the source.

•Results on time/cost reduction: The advantages mentioned around reduced time and production cost (line 182-186) would be strengthened by including results that specifically demonstrate this for the energy absorber.

•Traditional vs new design dimensions: Provide the dimensions of both the traditional design and the new design shown in Fig 3-6 for better comparison.

•CFRP modifications explained: Line 205-207 mentions modifications made, but a clearer explanation of the changes is needed.

•Image quality: Ensure all figures have a high enough resolution (dpi) for clear viewing.

Sincerely yours,

Author Response

Comment 1: CFRP modification: In the introduction, you mention CFRP but don't discuss the specifics of the modifications made.  A clearer explanation of the CFRP used in the energy absorber would be beneficial.

Response 1: Thank you for your comment. We acknowledge the need to clarify the specific modifications made to the CFRP used in our energy absorber. We have added a more detailed explanation in the introduction to elaborate on the exact nature of the modifications and their intended impact on the performance of the energy absorber.

Comment 2: Meaning of AB, ABA… type: Define the terminology used, such as the meaning of AB, ABA…  on line 62.

Response 2: Thank you for pointing this out. We agree that providing clear definitions for these terms would improve the reader’s understanding. We have now included definitions for AB, ABA, and ABC block copolymers in the introduction.

Comment 3: Energy absorber background: The introduction should provide more context on the energy absorber itself, including its applications. How does the autoclave process play a role in the functionality of the energy absorber?

Response 3: We appreciate your comment on the need for additional context regarding the energy absorber and its applications. We have expanded the introduction to provide a more comprehensive background on energy absorbers, their role in automotive applications, and the relevance of the autoclave and out-of-autoclave processes in their production.

Comment 4: Line 118-129 & Fig 1 clarity:  If the line setup described and Fig 1 represent a new process, a clear explanation is needed. If it references existing research, please cite the source.

Response 4: Thank you for your comment. Section 2.2 was enriched with more content. Already existing research reference has also been included.

Comment 5: Results on time/cost reduction: The advantages mentioned around reduced time and production cost (line 182-186) would be strengthened by including results that specifically demonstrate this for the energy absorber.

Response 5: Thank you for your comment. We have expanded our discussion to demonstrate the reductions in both time and cost achieved through our out-of-autoclave process for the energy absorber in 2.5 section.

Comment 6: Traditional vs new design dimensions: Provide the dimensions of both the traditional design and the new design shown in Fig 3-6 for better comparison.

Response 6: Thank you for your comment, however from the point of view of the design the overall dimensions do not change between autoclave (traditional) and compression moulding (process presented in the paper), only the process is different. Dimensions are as presented in Fig 4.

Comment 7: CFRP modifications explained: Line 205-207 mentions modifications made, but a clearer explanation of the changes is needed.

Response 7: Text has been adjusted to include the proper modifications. Electropolymerized CF fabric for our case.

Comment 8: Image quality: Ensure all figures have a high enough resolution (dpi) for clear viewing.

Response 8: Thank you for your comment. All figures have a high enough resolution. We have submitted them also in a separate file with the original uncompressed resolution.

Round 2

Reviewer 2 Report

Comments and Suggestions for Authors

Dear Authors,

In general, this version could be published, however, please check the figure quality, as well as the word size of all figures

Sincerely yours,

Author Response

Thank you for your comments. Figures have been sent into a seperate zip file to include them in the original quality.